



# Multiyear methane ebullition measurements from water and bare peat surfaces of a patterned boreal bog

Elisa Männistö[1], Aino Korrensalo[1], Pavel Alekseychik[2], Ivan Mammarella[2], Olli Peltola[2], Timo Vesala[2,3], Eeva-Stiina Tuittila[1]

[1]Peatland and soil ecology research group, School of Forest Sciences, University of Eastern Finland, PO Box 111, FIN-80101 Joensuu, Finland

[2]Institute for Atmospheric and Earth System Research/Physics, Faculty of Science, University of Helsinki, P.O. Box 68, 00014 Helsinki, Finland

[3]Institute for Atmospheric and Earth System Research/Forest Sciences, Faculty of Agriculture and Forestry, University of Helsinki, P.O. Box 27, 00014 Helsinki, Finland

*Correspondence to:* elisa.mannisto@uef.fi

**Abstract**

We measured methane ebullition from a patterned boreal bog situated in the Siikaneva wetland complex in southern Finland. Measurements were conducted on water (W) and bare peat surfaces (BP) in three growing seasons 2014–2016 using floating gas traps. The volume of the trapped gas was measured weekly, and methane and carbon dioxide ($CO_2$) concentrations of bubbles were analyzed from fresh bubble samples collected separately. We applied a mixed effects model to quantify the effect of the environmental controlling factors on the ebullition.

Ebullition was higher from W than from BP, and more bubbles were released from open water (OW) than from water's edge (EW). On average, ebullition rate was the highest in the wettest year 2016 and ranged between 0–253 mg m$^{-2}$d$^{-1}$, 0–147 mg m$^{-2}$d$^{-1}$ and 0–186 mg m$^{-2}$d$^{-1}$ in 2014, 2015 and 2016, respectively. Ebullition increased together with increasing peat temperature, weekly air temperature sum and atmospheric pressure, and decreasing water table (WT). Methane concentration in the bubbles released from W was 15–20 times higher and from BP 10 times higher than their $CO_2$ concentration. The proportion of ebullition fluxes upscaled to ecosystem level for the peak season was 2–8 % and 2–5 % of the total flux measured with eddy covariance technique and with chambers and gas traps, respectively. Thus, the contribution of methane ebullition from wet non-vegetated surfaces of the bog to the total ecosystem-scale methane emission appeared to be small.

**Keywords:** CH$_4$, ebullition, peatland, peat temperature, water table, atmospheric pressure



## 1 Introduction

Historically, bogs were commonly feared as people saw mysterious lights which gave rise to the tales of the "will o' the wisps" that lure travelers from their paths to sink into bog holes (Meredith, 2002). Nowadays, these lights are thought to be spontaneous combustion of peatland gases, such as methane, bubbling to the atmosphere, rather than deceptive fairies. However, the widespread folklore indicates that the phenomenon is well known around the world in peatland rich areas. Although currently peatlands are more known for their climate cooling impact as small carbon sinks and the storage of a third of the global soil carbon stock (Strack, 2008), they are also a major natural source of methane, a potent climate warming greenhouse gas (IPCC, 2014). The same high water table (WT) conditions that support accumulation of organic material as peat by slowing down aerobic decomposition also favor methane production by anaerobic microbes, methanogens (Archaea) (Hanson and Hanson, 1996). It has been predicted that carbon dioxide ($CO_2$) uptake typically offsets sustained methane emissions in natural ecosystems in the long term (i.e., several centuries), although with large spatiotemporal variability (Petrescu et al., 2015).

Methane is emitted from peatlands into the atmosphere via three routes: by diffusion from peat, transport through aerenchymatous vascular plants and by episodic bubble release i.e. ebullition (LeMer and Roger, 2001; Raghoebarsing et al., 2005). Large part of the produced methane is oxidized by methanotrophic bacteria in the aerobic peat layer above water level (Hanson and Hanson, 1996; LeMer and Roger, 2001; Larmola et al., 2010), and thus methane flux rate of a peatland depends on the rates of methane production and consumption, in addition to transportation within the peat to the atmosphere. As methane emitted through vascular plants or by ebullition bypasses the oxidation in the aerobic peat layer, these pathways can potentially release high amounts of methane into the atmosphere. Diffusion through peat and vascular plants have been regarded to be the dominant pathways of methane emissions and those emission pathways have been largely targeted with chamber measurements (e.g. Bubier et al. 2005; Ström et al., 2005; Turetsky et al., 2014). Alternatively, eddy covariance (EC) technique is used used to estimate the integrated ecosystem scale methane flux (e.g. Brown et al. 2014; Rinne et al. 2018), but are unable to differentiate the emission pathways.

Current models of global methane budget are still uncertain due to limited knowledge of the relative contribution of different factors controlling methane fluxes (Riley et al., 2011). The



largest source of uncertainty is the quantity of methane emissions from natural wetlands, such

as peatlands (Riley et al., 2011; Melton et al., 2013). Within peatland emissions, the largest uncertainty is related to the magnitude of ebullition (Peltola et al., 2018). We are aware of only few studies that have directly measured ebullition from boreal peatlands with gas traps. In the first one, Hamilton et al. (1994) carried out measurements over 24 hours and found no bubbles. In the three other studies conducted in a fen (Starck et al. 2005; Strack and Waddington 2008)

and a bog (Stamp et al. 2013) ebullition fluxes between $7 – 96$ mg $CH_4$ m$^{-2}$ d$^{-1}$ were detected but the importance of ebullition for the ecosystem flux remained unrevealed. Ebullition has also been measured in the field by separating peak methane releases from steady chamber flux (Riutta et al. 2007; Tokida et al. 2007; Goodrich et al. 2011) with emissions varying from $49–152$ mg $CH_4$ m$^{-2}$ d$^{-1}$ (Goodrich et al. 2011) to $48–1440$ mg $CH_4$ m$^{-2}$ d$^{-1}$ (Tokida et al. 2007).

These studies show contrasting results on the contribution of ebullition to the total emission. While Riutta et al. (2007) estimated the role of ebullition to be small in the two study years, Tokida et al. (2007) (with two sample plots) found that the proportion of ebullition may constitute up to 50% of the total flux. Results on mesocosm studies in laboratory conditions are similarly disparate as they show that the proportion of ebullition in the total emission varies

from 3% (Green and Baird 2013) to 50% (Christensen et al. 2013).

Similarly to chamber and EC measurements (Rinne et al. 2007; Jackowicz-Krczyński et al. 2010; Turetsky et al.; 2014, Mikhaylov et al., 2015; Rinne et al., 2018), direct ebullition studies have connected the rate of methane emission to peat temperature (Strack et al. 2005) relating to increasing microbial activity (Conrad et al. 1997). Noteworthly, incoming energy flux has

85 been shown to primarily control the methane production and ebullition in shallow subarctic lakes (Wik et al., 2014) that could be contrasted to peatland pools. Ebullition in peatlands has additionally been linked to decreasing WT and falling atmospheric pressure: the decrease of hydrostatic pressure increases the volume of the gas phase of methane in peat and releases it into the atmosphere (Tokida et al., 2007). Also, an increase in atmospheric pressure can trigger

ebullition by decreasing the bubble size due to compression and thus increasing the bubble mobility in shallow peat (Comas et al., 2011; Chen and Slater, 2015). Furthermore, peat structure has been shown to affect bubble sizes and determine whether ebullition is steady or erratic (Ramirez et al., 2016). However, the importance of these factors for ebullition is still based on the few studies, of which the longest covers two growing seasons (Strack and

Waddington 2008).



In this study, we measured methane ebullition from open water pools (W) and bare peat surfaces (BP) with gas traps in three subsequent growing seasons 2014 – 2016 in a boreal bog where methane fluxes were measured also with EC and static chambers techniques. We aimed to: (1.) quantify the spatial and temporal variation of methane ebullition from wet bog surfaces; (2.) study the controlling factors; and (3.) assess the contribution of ebullition from wet surfaces to the ecosystem level emission.

## 2 Materials and methods

The study was conducted in the ombrotrophic bog that is part of Siikaneva peatland complex situated in southern Finland (61°50'N, 24°12'E), 160 m a.s.l., within the southern boreal vegetation zone (Ahti et al., 1968). Annual rainfall of the area is 707 mm, the annual cumulative temperature is 1318 degree days, the average annual temperature is 4.2 °C and the average temperatures in January and July are -7.2 °C and 17.1 °C, respectively (30-year averages from the nearby Juupajoki-Hyytiälä weather station). The microtopography of the studied bog site varies from W and BP to hollows, lawns and hummocks. W and BP cover together approximately one fourth of the site (W 11.6% and BP 15.3% within 30 m radius from the EC tower of the site). The bottom layer is formed by *Sphagnum* mosses, except in W and BP that are devoid of moss. Sedges are the dominating vascular plants in hollows and lawns, whereas vascular plant vegetation on hummocks is dominated by dwarf shrubs. In BP, *Rhynchospora alba* is often the only plant species (Korrensalo et al. 2018a).

In order to measure methane ebullition from the studied bog, floating gas traps were placed in W and BP in three subsequent years 2014–2016. Only W and BP microfroms were chosen because we expected high ebullition from these waterlogged surfaces that have almost no vegetation, and because the sampling method required gas traps to be easily filled with water. The gas traps were constructed from inverted plastic funnels with diameters ranging between 14.3 cm and 24.5 cm (Fig. 1). A syringe with a three-way stopcock was attached to the narrow end of each funnel and the joint was covered with sealant to make it airtight. Piece of metallic netting coated with filter fabric was glued inside the funnels to prevent litter and small animals from entering the gas traps in the open water pools. The gas traps on W were attached to a floating styrofoam raft and placed in the pools in lines of two or three gas traps, anchored to the opposing shores of the pool with strings (Fig. 1). Some gas traps were anchored at the center of the pools (open water, OW), while the other gas traps were anchored at the water's edge



(EW). The gas traps on BP were placed next to boardwalks in the study site. The air was sucked out of the gas traps with an extra syringe until they were filled with water. The rate of

ebullition was measured weekly by sampling the gas volume that had replaced water in each gas trap.

16 gas traps were used (11 in W and 5 in BP) from 3 June to 25 September in 2014, 20 gas traps (13 in W and 7 in BP) from 13 May to 24 September in 2015, and 18 gas traps (12 in W and 6 in BP) from 27 May to 9 September in 2016.

Methane concentration of the gas caught in the traps was assumed to dilute during the weekly sampling periods due to diffusion, and therefore methane concentration of the bubbles was not measured from the weekly samples. Instead, the methane concentration of the releasing gas bubbles was measured by collecting fresh ebullition samples from W without disturbing the gas traps, and from BP that had no gas traps. Ebullition was triggered manually from the

sampled surfaces and the formed bubbles were caught in an extra gas trap, from where 20 ml samples were taken into vacuumed glass vials. The samples were analyzed with an Agilent Technologies HP 8690 gas chromatograph in Natural Resources Institute Finland (LUKE), Vantaa. Fresh ebullition samples were collected four times during the measurement season in 2014 and 2016, and 13 times in 2015. Average methane concentration was interpolated from

the fresh ebullition samples for each weekly measurement period.

Average methane emission by ebullition as ml m$^{-2}$ d$^{-1}$ was calculated based on the area of the gas trap, number of days and volume of gas collected in each measurement period and the average methane concentration of each measurement period. In order to convert the emissions to mg m$^{-2}$ d$^{-1}$, methane density in each measurement period was calculated based on the average

air temperature of the measurement period in °C and the standard atmospheric air pressure, 1.01325 bar. Average methane emission (mg m$^{-2}$ d$^{-1}$) was calculated separately for ebullition from OW, EW and BP.

In order to compare the ebullition fluxes to EC and chamber measurements (Korrensalo et al. 2018b), the ebullition flux was upscaled to ecosystem level by interpolating the total average

ebullition that was calculated as a sum of average ebullition fluxes from W and BP weighted with their relative surface areas.

Air pressure and temperature data from 2014–2016 were received from the Juupajoki-Hyytiälä weather station that is situated about 6 km from the study site in Siikaneva. The data on WT, water temperature and peat temperatures at the depths of 5, 20 and 50 cm were received from





a data logger installed in the study site. Photosynthetically active radiation (PAR) data was measured in the site.

Linear mixed-effects models were used to analyze the effect of measured environmental variables (peat temperature in different depths, WT, atmospheric pressure and cumulative PAR and effective temperature sum of a measurement period as variables of incoming energy flux)

on log-transformed ebullition flux rates. The measurement funnel was included as a random effect in the model. We also tested which of the four peat temperature variables explained the variation in ebullition fluxes the best. The data were analyzed with the function lme of the package nlme of the R software (version 3.3.2).

**3 Results**

Among the three studied years, the year 2014 was the warmest, driest and with the highest amount of cumulative photosynthetically active radiation (PAR) (Finnish Meteorological Institute open data) (Table 1). It was also warmer than 30y average. Year 2015 was the coolest, with a lowered annual rainfall and PAR, while 2016 was the wettest and the cloudiest year

(Table 1). All three years were significantly drier than the average (Table 1).

Measured methane ebullition ranges were 0–253 mg m$^{-2}$d$^{-1}$, 0–147 mg m$^{-2}$d$^{-1}$ and 0–186 mg m$^{-2}$d$^{-1}$ in 2014, 2015 and 2016, respectively (A1). When ebullition was averaged between all the funnels for each measurement period the total weekly means were 1–90 mg m$^{-2}$d$^{-1}$ in 2014, 0–41 mg m$^{-2}$d$^{-1}$ in 2015 and 14–71 mg m$^{-2}$d$^{-1}$ in 2016. The three years differed (degrees of

freedom (DF) = 2, 746; $p$ <.0001) as slightly higher ebullition fluxes were generally obtained in 2015 than 2014, while, on average, the ebullition fluxes were on their highest in the wettest year 2016.

Higher ebullition was observed on W than on BP (Fig. 2) (DF = 1, 746; $p$ <.0001). Ebullition from OW was significantly higher than ebullition from EW, except in the middle of the

growing season 2015 (Fig. 2). Although BP showed lower ebullition with fewer peaks than W, all the surfaces had the same seasonal ebullition pattern each year with highest fluxes observed in August (Fig. 2). However, in 2015 the highest ebullition was measured later than in other years after relatively low ebullition in late summer (Fig. 2).

Ebullition increased with increasing average peat temperature at the depth of 5 cm (DF = 1,

746; $p$ <.0001) that explained ebullition better than the other peat temperature variables



measured. Seasonal pattern of ebullition followed temperature in each year (Fig. 3). Higher ebullition rates were also explained with decreasing average WT (DF = 1, 746; $p$ = 0.0001). The highest ebullition peaks were associated with the lowest WT in each year (Fig. 3). A prolonged depression of WT further explained the late peak of ebullition in 2015, as well as the increase in ebullition in the autumn 2016 (Fig. 3). Change in atmospheric pressure during the measurement period further explained ebullition: more bubbles were released with higher increase in pressure (DF = 1, 746; p = 0.001). Some events of ebullition might be directly related to decreasing atmospheric pressure such as the small peak in ebullition in mid-August 2014 that appears to be better explained by long decrease in atmospheric pressure than by peat temperature or WT (Fig.4). After including peat temperature, WT and change in atmospheric pressure, the effective temperature sum of a measurement period still had a positive effect on ebullition (DF = 1, 746; $p$ = 0.0351. Finally, the cumulative PAR had no significant effect on ebullition and was excluded from the final model.

Fresh ebullition samples analyses showed that the released gas bubbles contained more methane than $CO_2$. Methane concentration of bubbles released from W was 15–20 times higher than their $CO_2$ concentration, while bubbles from BP had tenfold higher methane than $CO_2$ concentration (Table 2).

The average ebullition flux upscaled to ecosystem level was an order of a magnitude lower than the net methane flux measured by EC in each year (Fig. 5). The sum of ebullition and upscaled chamber flux in 2014 was higher than the one measured with EC, but the two estimates followed the same seasonal trend (Fig. 5). The contribution of ebullition to the total methane flux measured with chambers and bubble traps during the peak season in 2014 was 2, 3 and 5 % in June, July and August, respectively (Table 3). The contribution of ebullition to EC flux during the peak season varied from 2 % in June 2014 to 8 % in August 2015 (Table 3).

## 4 Discussion

### 4.1 The magnitude of ebullition

The methane ebullition measured in this study ranged from 0 to 253 mg m$^{-2}$ d$^{-1}$ and the seasonal weekly mean ebullition for different surfaces ranged from 3 (measured from BP in 2014) to 47 mg m$^{-2}$ d$^{-1}$ (measured from OW in 2016). Our results are of the same magnitude as ebullition





fluxes previously measured in boreal peatlands with gas traps, ranging from 7 to 96 mg m$^{-2}$ d$^{-1}$ (Strack et al., 2005; Strack and Waddignton, 2008; Stamp et al., 2013), and with automatic chambers, ranging from 9 to 152 mg m$^{-2}$ d$^{-1}$ (Goodrich et al., 2011). In addition to field

measurements some of the laboratory-based experiments have shown similar ebullition flux rates in the range of 0-270 mg m$^{-2}$ d$^{-1}$ (Christensen et al., 2004; Kellner et al., 2006; Yu et al., 2014) but also higher fluxes up to 784 mg m$^{-2}$ d$^{-1}$ (Green and Baird, 2012). Some laboratory studies have shown potential for even much higher ebullition rates up to 33 000 mg m$^{-2}$ d$^{-1}$ (*Sphagnum* surface samples from bog in Tokida et al., 2005; fen lawn samples in Waddington

et al. 2009). So far, only Tokida et al. (2007) have estimated ebullition fluxes reaching 1440 mg m$^{-2}$ d$^{-1}$ in the field based on methane fluxes measured with the static chamber method from two sample plots showing high episodic fluxes during 30 min measurements. Generally, there is a difference in temporal resolution between the two methods as chamber measurements usually cover only short time periods (from minutes to hours) while gas traps show estimates

of cumulative bubble flux over several days.

The fact that the ebullition rates measured with gas traps are lower than in laboratory studies might be partly explained by the process of bubbles stacking in the funnels instead of automatically gathering in the headspace. In this study, we tried to overcome this potential error source by gently shaking and tapping the funnels before sampling, simultaneously trying

to avoid causing more ebullition by this disturbance. However, methane ebullition fluxes of up to 1683 mg m$^{-2}$ d$^{-1}$ have been previously measured with the same method from subarctic lakes (Wik et al. 2013), which shows the potential of this method to measure also higher ebullition fluxes.

**4.2 Temporal and spatial variation**

Our study conducted over three growing seasons showed interannual variation. The highest ebullition on average was measured in 2016, whereas the average flux rates of 2014 and 2015 did not differ significantly from each other. More ebullition was measured especially from BP in 2016 that was the wettest year with the highest WT. This indicates that despite higher WT

increases hydrostatic pressure in peat, wetter conditions in BP facilitate gas release as bubbles. Although 2015 was almost as wet year as 2016, it was much cooler, which decreases methane production. The warmest year 2014 again was much drier than 2016, and although there was high ebullition with sharp drop in WT during the peak season, the general ebullition level from



BP was low. The only other peatland study with gas traps covering more than one growing
season (Strack and Waddington 2008) also found the ebullition level to differ between the study
years. Similarly, Wik et al. (2013) found differences in bubble methane concentrations and
fluxes in subarctic lakes among the four summers studied. These results point out the need for
multi-year studies in order to include inter annual variation of ebullition fluxes in methane
models. Furthermore, the higher ebullition rate from W than from BP in our study indicates
that balanced sampling in a bog should cover microform variability, although in some studies
no spatial variation in ebullition were found (Green and Baird, 2012 and 2013; Stamp et al.,
2013). However, drier and wetter conditions can change the proportions of water and bare peat
surfaces, and according to our results, such changes may have an impact on ebullition.

### 4.3 Controlling factors and their importance

The measured ebullition rates increased together with peat temperature as shown also earlier
(Strack and Waddington, 2008). Increasing temperature generally increases the activity of
methanogens, and thus more methane is produced in the peat when it gets warmer until the
temperature optimum of the microbes around 20-30 °C is reached (Dunfield et al., 1993). Peat
temperature affects also the solubility of methane according to Henry's law as gas solubility
decreases with increasing temperature (Strack 2005). Thus, increasing peat temperature may
lead to transfer of methane from aqueous to gaseous phase, which increases bubble formation
(Strack, 2005). In our study, the peat temperature at the depth of 5 cm showed the highest
correlation with ebullition but temperature in all depths were highly intercorrelated. The effect
of peat temperature was reflected in the seasonal pattern of ebullition.

As expected, ebullition fluxes increased when WT decreased as found in previous studies
(Strack and Waddington, 2008). Bubbles may accumulate in peat under barriers, such as pieces
of wood, and they are suppressed by high hydrostatic or air pressure (Rosenberry et al., 2003;
Strack and Waddington, 2008; Chen and Slater, 2015). Decreasing WT lowers the hydrostatic
pressure releasing newly formed and the accumulated bubbles. Many studies have also shown
that the falling atmospheric pressure can trigger high rates of ebullition (Tokida et al. 2005;
Tokida et al., 2007). Although some weeks showed higher ebullition rates when atmospheric
pressure was falling, this pattern was not consistent as increasing ebullition rates were
measured also during periods of rising atmospheric pressure. After including WT as
explanatory variable, we still found the weekly change in atmospheric pressure to significantly





affect ebullition as bigger increase in weekly pressure was related to more ebullition. Previously, Comas et al. (2011) used ground penetrating radar (GPR) to study the vertical distribution of free-phase gas in a northern peatland and found that increasing atmospheric pressure caused rapid ebullition by releasing gas from shallow peat, whereas decreasing

pressure released gas from deeper peat to shallow layers. Also Chen and Slater (2015) showed that increasing pressure can trigger ebullition as it increases the bubble mobility in peat.

Furthermore, higher ebullition rates were measured with higher effective temperature sum of the measurement period. This indicates the importance of energy input as a driver of methane production and release as shown by Wik et al. (2014). They found strong positive correlations

between seasonal bubble methane flux from subarctic lakes and four proxies of energy flux, such as average short wave radiation and maximum water sediment temperature (Wik et al., 2014). We tried also to compare cumulative PAR (i.e. short wave radiation) to seasonal cumulative ebullition fluxes but could not find clear correlation between the two in the three study years. However, the positive effect of the measurement period temperature sum on

ebullition shows that increasing energy input can increase ebullitive methane flux rates in the studied bog site.

## 4.4 Importance for the ecosystem level flux

When measured ebullition fluxes were upscaled to the ecosystem level, they showed much

lower methane emissions than measured with chamber and EC techniques. In our previous study (Korrensalo et al., 2018b), we measured diffusive methane fluxes with the static chamber technique from six different plant community types, including BP, at the same bog site in 2014. We found higher methane fluxes from BP than from high hummocks (HHU), but otherwise all the studied plant community types had similar methane fluxes. When chamber fluxes were

upscaled to ecosystem level they were similar to the EC flux (Korrensalo et al., 2018b). Although laboratory incubation studies have shown that the contribution of ebullition to the total methane flux may reach up to 50 % (Christensen et al., 2013; Tokida et al., 2007), the ebullition contribution in this study was only 3-5 % during the peak season of 2014. Here, ebullition is considered only from waterlogged surfaces as we did not measure ebullition from

vegetated surfaces. Previously, Riutta et al. (2007) measured methane fluxes from different plant communitites with static chambers in Siikaneva fen site situated 1.3 km south-east from our studied bog site and calculated results for both diffusive and ebullition fluxes. They found



ebullition from all communities but showed that its contribution to the total flux (diffusive flux + ebullition) was negligible or very small (Riutta et al., 2007). Therefore, we assume that

ebullition from vegetated surfaces would not greatly contribute to the total flux in the bog site either. Earlier, similarly to our study, Green and Baird (2013) have found ebullition to contribute less than 3.3 % of total methane fluxes when incubating peat samples collected from hollows and lawns from two raised bogs in laboratory study. As the measured bubble methane fluxes in our study were of the same magnitude in each year, ebullition did not contribute

significantly to the ecosystem methane emissions in any studied growing season as seen in the comparison with the EC flux. While the same seasonal trend and peaks can be seen in both fluxes in each year, the total flux measured with EC is constantly at least an order of magnitude higher than the ebullition flux rate.

**Conclusions**

More methane ebullition was found from W than from BP, and within the pools more bubbles were released from OW than from EW. We found also variation between the three studied growing seasons as ebullition rate was generally higher in the wettest year 2016. Due to this spatial and temporal variation, differences between years in wet/dry conditons may have an

effect on ebullition. As expected, ebullition increased together with increasing peat temperature, that facilitates methane production, and with decreasing WT, that reduces hydrostatic pressure on peat. Additionally, more bubbles were released with bigger weekly increase in atmospheric pressure, which has been related to rapid ebullition from shallow peat. Futhermore, higher weekly temperature sum had a positive effect on ebullition, which shows

that increasing energy input can increase ebullitive methane flux rates in the studied bog site. Therefore, the growing season lengthening and increase in the average temperatures due to climate change may increase the methane emissions in the peatland ecosystem as long as waterlogged anoxic conditions in the peat for methane production persist. Ebulliton flux upscaled to the ecosystem level showed similar seasonal pattern as methane fluxes measured

with EC and chamber techniques but was an order of magnitude lower and had a very small contribution to the total ecosystem flux. Our study includes ebullition only from the waterlogged surfaces as we did not expect ebullition from all the plant community types to be substantial based on the previous study in the nearby fen site. However, estimating the amount of ebullition from all the plant community types would be needed to fully understand the spatial



variation of ebullition in the future. In addition, measurements with e.g. time-lapse cameras are needed to study the short term temporal variation of ebullition and to estimate the frequency and magnitude of rapid ebullition events that may contribute to the total ecosystem flux.

**Data availability**

Data is available upon request from the corresponding author.

**Author contribution**

AK, EST, PA and TV came up with the idea and design. AK and EM conducted the ebullition measurements and processed the data. Eddy covariance data was collected and analyzed by PA and IM. EM fitted the mixed-effects models. The manuscript was written by EM, AK and EST and commented by all the other authors.

**Competing interests**

The authors declare that they have no conflict of interest.

**Acknowledgements**

This work is supported by faculty of Science and Forestry, University of Eastern Finland, the Finnish Cultural Foundation, Academy of Finland (project code 287039), Academy research
project CLIMOSS (41007-00086900), Strategic Research Council research project SOMPA (41007-00114600), the National Centre of Excellence (272041), ICOS-Finland (281255), Academy professor project (284701) funded by the Academy of Finland and AtMath funded by the University of Helsinki. OP was supported by the Postdoctoral Researcher project (315424) funded by the Academy of Finland. We thank Hyytiälä Forest Research Station and
its staff for research facilities and Pauli Karppinen from Natural Resources Institute Finland (LUKE) for analyzing methane concentration of the fresh bubble samples. We also thank Salli Uljas, Janne Sormunen, Franziska Rossocha, and Laura Kettunen for the help in the field.

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



**Tables**

Table 1. Effective temperature sum of the growing season, annual rainfall and the amount of
photosynthetically active radiation (PAR) in the three studied years 2014–2016 and compared
to the 30 years averages of the area. Data for the Hyytiälä weather station from Finnish
Meteorological Institute open data.

| Year | Temp. sum degree days | Annual rainfall mm | PAR µmol m$^{-2}$ |
|---|---|---|---|
| 2014 | 1 349 | 579 | 70 800 |
| 2015 | 1 166 | 658 | 69 180 |
| 2016 | 1 280 | 660 | 67 996 |
| 30 year mean | 1 318 | 707 | - |






Table 2. Average methane ($CH_4$) and carbon dioxide ($CO_2$) concentrations (ml/l) with standard deviation (SD) of releasing gas bubbles from pools (W) and bare peat surfaces (BP) in the three studied years 2014–2016.

| | 2014 | | 2015 | | 2016 | |
|---|---|---|---|---|---|---|
| | Average | SD | Average | SD | Average | SD |
| W $CH_4$ | 380.0 | 50.7 | 285.0 | 93.0 | 423.0 | 103.9 |
| W $CO_2$ | 23.8 | 4.9 | 18.0 | 8.8 | 20.8 | 6.5 |
| BP $CH_4$ | 274.2 | 64.5 | 273.6 | 56.1 | 364.2 | 123.7 |
| BP $CO_2$ | 29.4 | 16.0 | 26.9 | 9.4 | 31.8 | 8.9 |







Table 3. Monthly cumulative methane fluxes (mg m$^{-2}$ mo$^{-1}$) measured as ebullition and with eddy covariance (EC) technique for June–August in the three studied years 2014–2016.

|  | 2014 | | | 2015 | | | 2016 | | |
|---|---|---|---|---|---|---|---|---|---|
|  | Ebullition | EC | % of ebullition | Ebullition | EC | % of ebullition | Ebullition | EC | % of ebullition |
| June | 27 | 1668 | 2 | 73 | 1139 | 6 | 117 | 2530 | 5 |
| July | 155 | 3423 | 5 | 112 | 2277 | 5 | 314 | 4216 | 7 |
| August | 176 | 3447 | 5 | 223 | 2657 | 8 | 249 | 3448 | 7 |







**Figures**

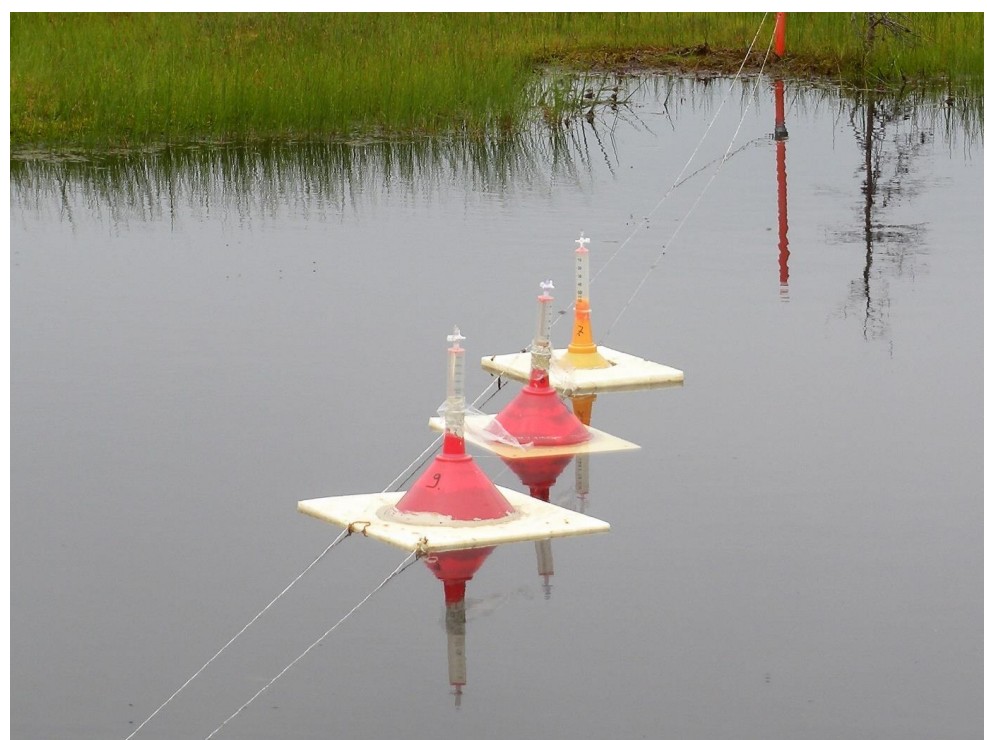

Figure 1. Floating gas traps in open water peatland pool (OW).





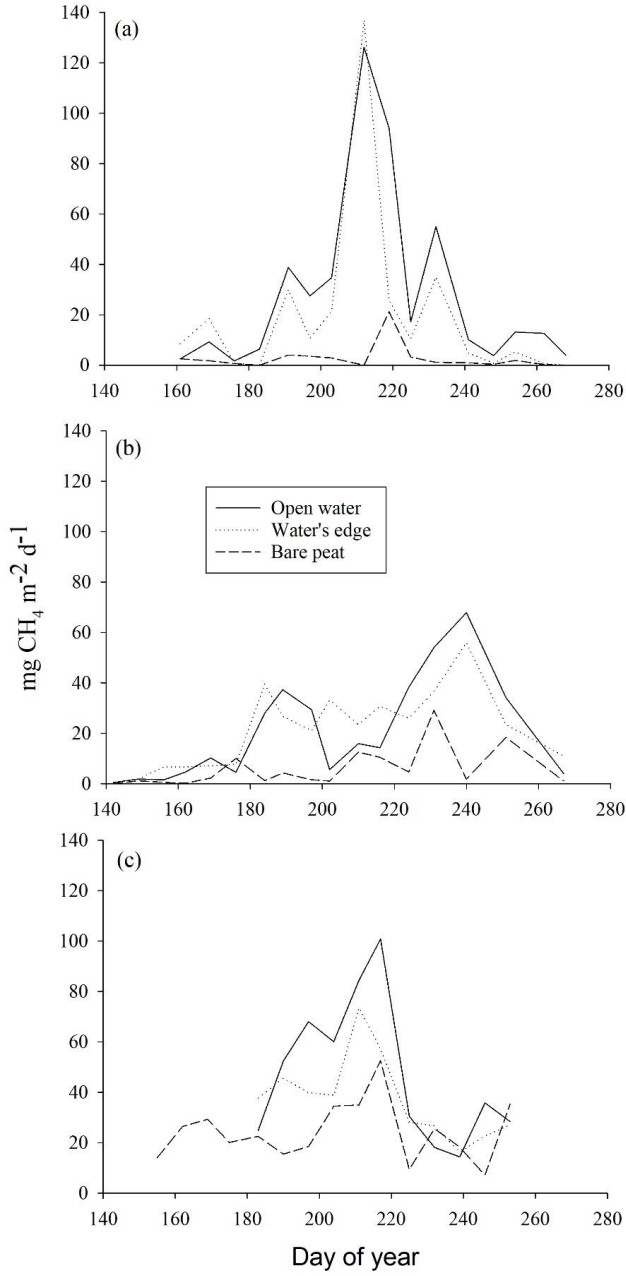

Figure 2. Mean methane ebullition measured weekly in Siikaneva bog in three consecutive
years a) 2014, b) 2015 and c) 2016 over different surfaces: bare peat surfaces, open water and
water's edge of pools.

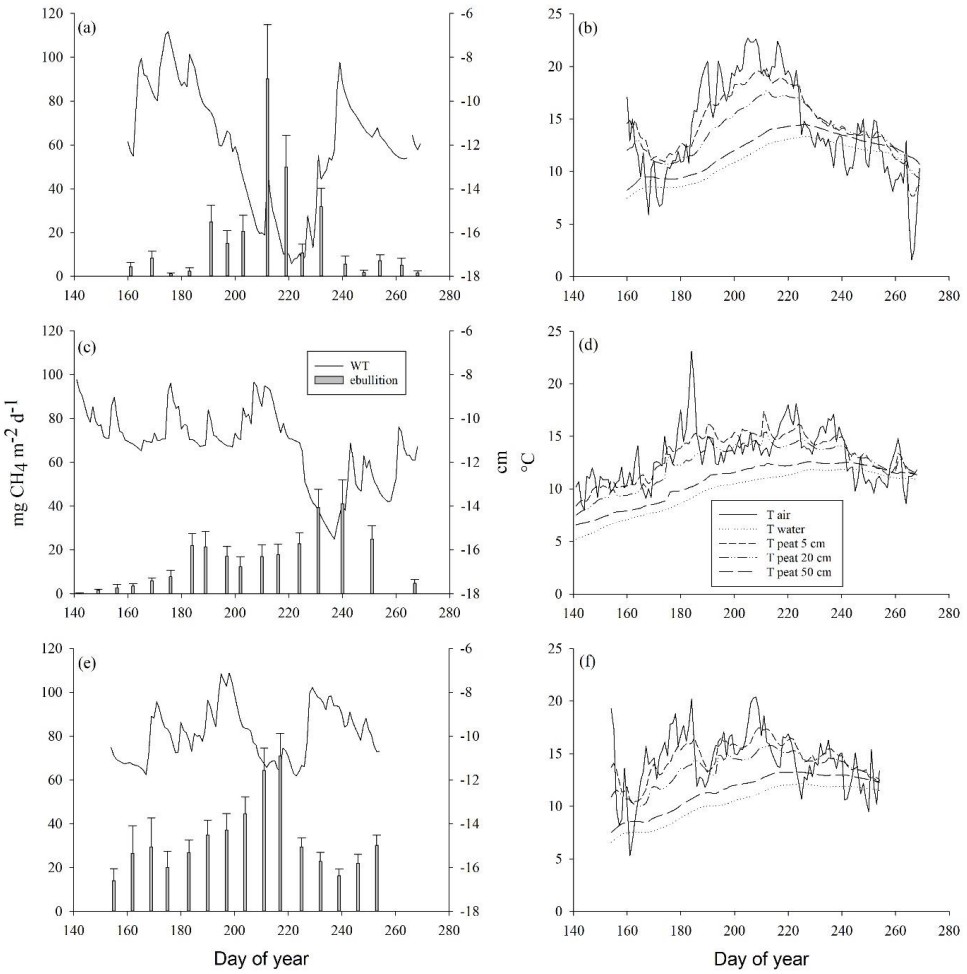

Figure 3. Mean weekly methane ebullition with standard error of the means from all surfaces compared to water table (WT) (left panel), and air, water and peat temperatures in the depths of 5, 25 and 50 cm (right panel) measured in Siikaneva bog in years a-b) 2014, c-d) 2015 and e-f) 2016.



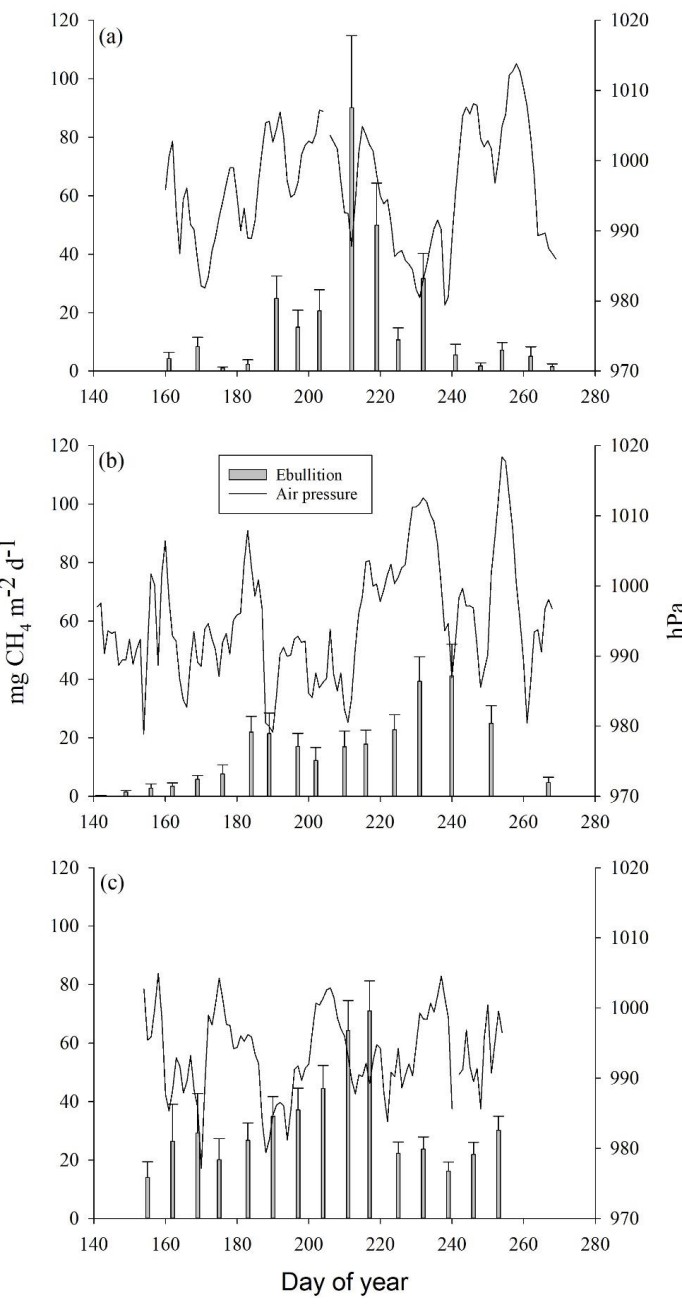

Figure 4. Mean weekly methane ebullition from all surfaces compared to atmospheric pressure measured in Siikaneva bog in a) 2014, b) 2015 and c) 2016.



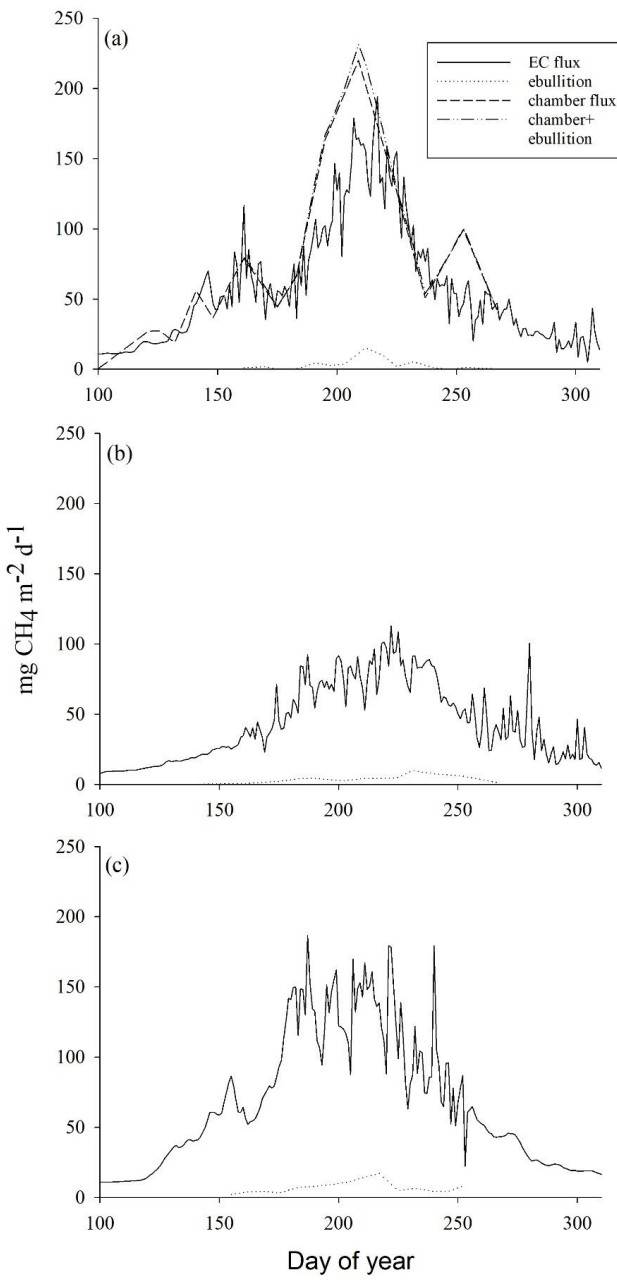


Figure 5. Ecosystem level methane fluxes measured with the eddy covariance (EC) technique and upscaled from ebullition measurements in a) 2014, b) 2015 and c) 2016. In 2014, ecosystem level methane fluxes are compared also to upscaled chamber fluxes.



**Appendix A.**

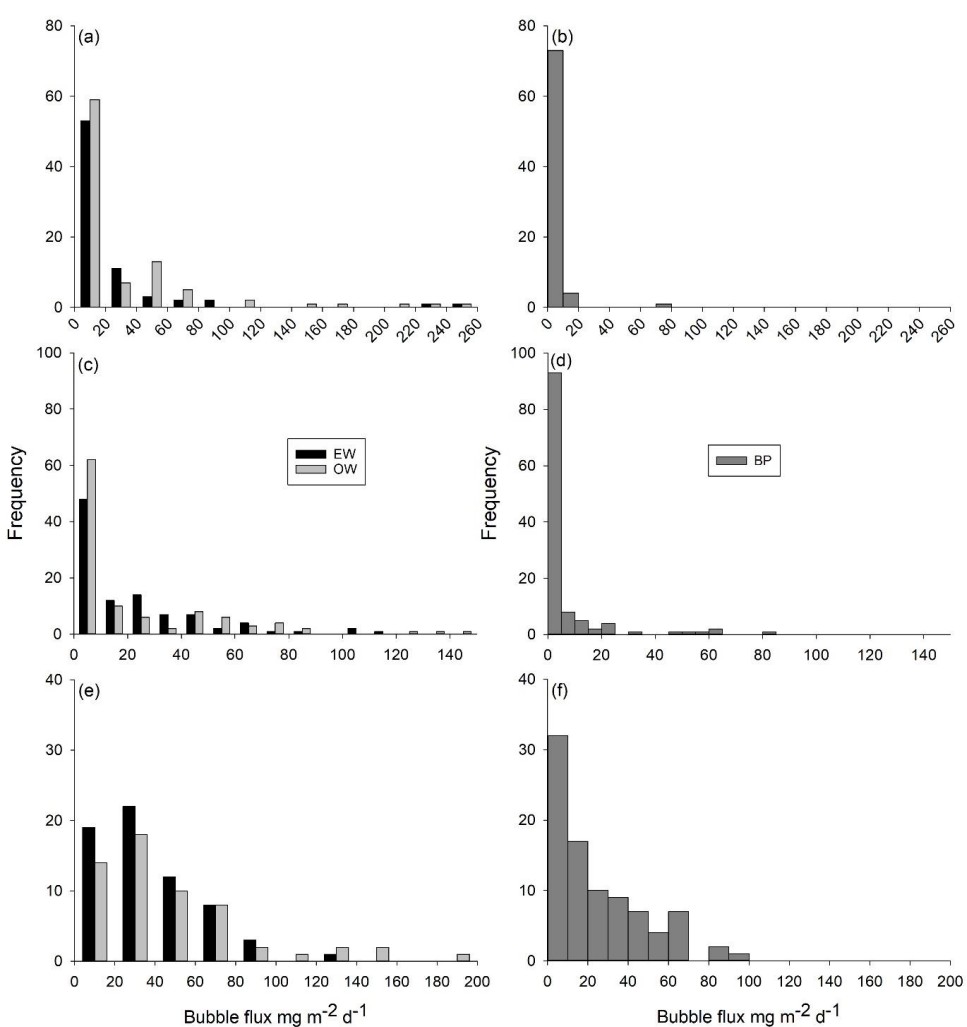

Figure A1. Frequency distribution of methane ebullition (mg m$^{-2}$ d$^{-1}$) per gas trap from open water pools (OW), water's edge (EW) and bare peat surfaces (BP) in a-b) 2014, c-d) 2015 and e-f) 2016. Note differences in scales between the years.