# Peer review of "Multiyear methane ebullition measurements from water and bare peat surfaces of a patterned boreal bog"

_Biogeosciences, 2018_

## Referee Comment (RC1) · Anonymous Referee #1 · 13 Feb 2019

Report on manuscript titled "Multiyear methane ebullition measurements from water and bare peat surfaces of a patterned boreal bog".

General comments

I found this paper well written and containing useful results. Design of the study is optimal for chosen object and helps to represent studied phenomena. Methods were described good enough. Especially I appreciate honest remarks on several methodological details. Literature was presented comprehensively, all important available studies were cited and used for comparison. It is very nice that photo of studied object was given in the paper (but in comments below I strongly recommend to give more photos to give a reader as much as possible information on your object). Nevertheless I recommend to improve this manuscript in several directions (they are described in specific comments section). After that I think this paper should be published.

Specific comments

1. Lines 54-55. It seems to me that in these lines it was pointed out that during methane transport in vascular plants methane is not consumed by methanotrophs. I think this is wrong because there are a lot of papers showing presence of methanotrophs in plant tissues (for example, Bao et al., 2014; Doronina et al., 2004; King, 1994). Am I right?

2. Lines 110-115. I think your object should be described in paper text more comprehensively. I understand that there are a lot of papers about Siikaneva station peatland. But more information important for YOUR study should be given. In general you describe your research sites as peatland. But you also compare methane fluxes from your site with fluxes from ponds and lakes. It seems to me that there is an ambiguity. Do you think that your objects are something between shallow lake and peatland?

It is very important for future reader to understand what EXATLY is your object. That is why more information on factors of methane emission should be given in the paper text. What was the water depth in all three site types (OW, EW and BP)? Why bare peat is bare? It is not typical for natural intact wetlands in Canada or Russia, where water table depth is about 0 cm or higher and moss cover is continuous. Why these parts of peatland are so wet (or so submerged)? Does peat on your sites removed by erosion (or by any secondary process, but not in inherent peatland development)? If so it decreases methane emission (because relatively young and rich in substrates peat layer was removed). Or your sites are in an inner water channel (stream) inside the peatland?

I strongly recommend to add photos of bare peat surface and water's edge (EW in your terms) sites and probably small map of your peatland to see where your sites are situated.

3. Lines 127-128. How do you define where is open water (OW) and where is water's edge (EW)? Based on water depth? It is important because methane emission is known to be WT-dependent (see Wik's papers from your reference list for example).

4. Lines 137-139. I have several questions to discuss on using triggered ebullition gas concentration for calculation of the total emission.

a) As I understand bubbles release to the surface from the peat when methane concentration in them reaches a certain threshold. If it is right, triggered ebullition gas concentration is lower than "real" ebullition gas concentration because gas concentration in triggered bubbles did not reach a certain threshold. Hence methane and $CO_2$ concentrations and their fluxes are underestimated using your methodology (not by much I think). If I am right please mention it in the paper text.

b) Methane is poorly dissolved in water. And concentration of dissolved methane in peatland water is usually high and close to saturation level. That's why I think that methane concentration in funnels during a week (actually it is less than a week, because bubbles do not release right after funnel installation) can be more or less constant and not decreased by diffusion. Did your compare methane concentration in funnels after week of exposition and methane concentration after triggered ebullition in the end of this week? I have never read about methodology you used and think that it is novel. Any novel methodology should be assessed. For example, Martin Wik (see papers cited in your manuscript) use methane concentration in funnels after week (or couple of weeks) of exposition. It is always risky to use not *in situ* concentration in such heterogeneous environments as peatlands.

5. Line 158. Where exactly your water table sensor was placed, in what site? As I see on Figure 3 you use the same WT data for prediction of fluxes from all three types of sites. But I think it is not correct because each site type has own WT mean level and WT seasonal dynamic. Anyway it must be mentioned in a paper text.

Bao, Z., Okubo, T., Kubota, K., Kasahara, Y., Tsurumaru, H., Anda, M., ... & Minamisawa, K. (2014). Metaproteomic Identification of Diazotrophic Methanotrophs, and their Tissue Localization in Field-grown Rice Roots. Applied and environmental microbiology, AEM-00969.

Doronina, N. V., Ivanova, E. G., Suzina, N. E., & Trotsenko, Y. A. (2004). Methanotrophs and methylobacteria are found in woody plant tissues within the winter period. Microbiology, 73(6), 702-709.

King, G. M. (1994). Associations of methanotrophs with the roots and rhizomes of aquatic vegetation. Applied and Environmental Microbiology, 60(9), 3220-3227.

---

## Author Comment (AC1) · 11 Mar 2019

Dear referee #1,

Thank you for the constructive comments that will help us to improve the manuscript. Please find our responses below. We will revise the manuscript accordingly after the review process is closed by the editor; meanwhile we would be happy to get your response for the few cases where we disagree.

**Referee #1:**

General comments

I found this paper well written and containing useful results. Design of the study is optimal for chosen object and helps to represent studied phenomena. Methods were described good enough. Especially I appreciate honest remarks on several methodological details. Literature was presented comprehensively, all important available studies were cited and used for comparison. It is very nice that photo of studied object was given in the paper (but in comments below I strongly recommend to give more photos to give a reader as much as possible information on your object). Nevertheless I recommend to improve this manuscript in several directions (they are described in specific comments section). After that I think this paper should be published.

Specific comments

**Comment:** Lines 54-55. It seems to me that in these lines it was pointed out that during methane transport in vascular plants methane is not consumed by methanotrophs. I think this is wrong because there are a lot of papers showing presence of methanotrophs in plant tissues (for example, Bao et al., 2014; Doronina et al., 2004; King, 1994). Am I right?

**Response:** Yes, this is true for several wetland species such as rice *Oryza sativa* (Bosse and Frenzel, 1997) and cattail *Typha latifolia* and *Calamagrostis canadensis* (King, 1994). However, in bogs so far significant methane oxidation has not been detected in vascular plants, such as *Eriophorum angustifolium* and *E. vaginatum* (Frenzel and Rudolph, 1998), but in *Sphagnum* mosses (Larmola et al. 2010). We will included this in the Introduction.

**Comment:** Lines 110-115. I think your object should be described in paper text more comprehensively. I understand that there are a lot of papers about Siikaneva station peatland. But more information important for YOUR study should be given. In general you describe your research sites as peatland. But you also compare methane fluxes from your site with fluxes from ponds and lakes. It seems to me that there is an ambiguity. Do you think that your objects are something between shallow lake and peatland?

It is very important for future reader to understand what EXATLY is your object. That is why more information on factors of methane emission should be given in the paper text. What was the water depth in all three site types (OW, EW and BP)? Why bare peat is bare? It is not typical for natural intact wetlands in Canada or Russia, where water table depth is about 0 cm or higher and moss cover is continuous. Why these parts of peatland are so wet (or so submerged)? Does peat on your sites removed by erosion (or by any secondary process, but not in inherent peatland development)? If so it decreases methane emission (because

relatively young and rich in substrates peat layer was removed). Or your sites are in an inner water channel (stream) inside the peatland?

I strongly recommend to add photos of bare peat surface and water's edge (EW in your terms) sites and probably small map of your peatland to see where your sites are situated.

**Response:** Our study site is a patterned peatland with parches without moss cover. Those patches are either shallow with visible peat surface having their water level at the surface or deeper, covered by open water. In a way, our objects are something between very small and shallow lake and peatland.

Water depth in BP in 2014 was on average -1.8 cm. Water depth in the pools is not easy to measure as it is hard to determine what is the bottom of the pools: on average there is one meter of water over very loose peat slurry. The water depth does not generally differ between EW and OW as these peatland pools do not have shallower banks, but are deep right away at the edge of the surrounding moss cover. The shallow arctic lakes in Wik et al. (2013) have organic sediment layers and their water depth varies around 0-6 m making them not that different from our pools except being much larger in their area.

The bare peat surfaces or 'mud-bottom hollows' found in our site are commonly described in boreal peatlands in, Estonia, Finland, Sweden, Russia, Western Europe and North America (Karofeld et al., 2015 and the references therein).  Their formation processes are not well known but according to Karofeld et al. (2015) at least in this region they are not formed due to erosion but are inherent part of peatland development.

We will include a more detailed description of our study objectives and the aerial photo of the site (Figure 1. below) to Methods.

[Figure]

Figure 1. Aerial photo of the study site in Siikaneva bog. Red lines with dots mark the floating gas traps in open water (OW) and water's edge (EW). Red circles mark the area within what the gas traps were placed on bare peat surfaces (BP) that are seen as brownish-grey in the photo. The eddy covariance (EC) raft is marked with the red x.

**Comment:** Lines 127-128. How do you define where is open water (OW) and where is water's edge (EW)? Based on water depth? It is important because methane emission is known to be WT-dependent (see Wik's papers from your reference list for example).

**Response:** EW and OW were not defined based on the water depth but simply by the distance to the nearest vegetated surface. EW was right next to the moss and OW was the furthest away from it in the middle of the pool (at least 1m, on average about 3 m away from the edges). This separation was chosen to reflect the potential difference in availability of substrate for methanogenesis.

**Comment:** Lines 137-139. I have several questions to discuss on using triggered ebullition gas concentration for calculation of the total emission.

**a)** As I understand bubbles release to the surface from the peat when methane concentration in them reaches a certain threshold. If it is right, triggered ebullition gas concentration is lower than "real" ebullition gas concentration because gas concentration in triggered bubbles did not reach a certain threshold. Hence methane and CO2 concentrations and their fluxes are

underestimated using your methodology (not by much I think). If I am right please mention it in the paper text.

**b)** Methane is poorly dissolved in water. And concentration of dissolved methane in peatland water is usually high and close to saturation level. That's why I think that methane concentration in funnels during a week (actually it is less than a week, because bubbles do not release right after funnel installation) can be more or less constant and not decreased by diffusion. Did your compare methane concentration in funnels after week of exposition and methane concentration after triggered ebullition in the end of this week? I have never read about methodology you used and think that it is novel. Any novel methodology should be assessed. For example, Martin Wik (see papers cited in your manuscript) use methane concentration in funnels after week (or couple of weeks) of exposition. It is always risky to use not in situ concentration in such heterogeneous environments as peatlands.

**Response:**

**a)** Bubbles are not released from the peat after reaching certain methane concentration but depending on the bubble volume (Peltola et al., 2018). Volume in turn is controlled by pressure and temperature that affect methane solubility.

**b)** We have measured methane concentration of the gas caught in the funnels in 2014 and compared those with the fresh samples triggered from the peat. The concentrations in the funnels were clearly lower than in the fresh ebullition samples (see the table below). Therefore, after 2014 we did not measure concentrations from the funnels anymore but only took fresh bubble samples to get more accurate concentrations. We will include this in the methods; Table could be added as an appendix.

| Date | Mean CH4 concentration ml/l | | | |
| --- | --- | --- | --- | --- |
| | W funnel | BP funnel | W fresh | BP fresh |
| 10-Jun | 17 | 45 | 293 | 268 |
| 16-Jul | 61 | *245* | 420 | 313 |
| 13-Aug | 86 | 152 | 399 | 172 |
| 05-Sep | 49 | *63* | 379 | 221 |

*Numbers in Italic indicate the concentration in a single measured funnel, not a mean of many funnels*

**Comment:** Line 158. Where exactly your water table sensor was placed, in what site? As I see on Figure 3 you use the same WT data for prediction of fluxes from all three types of sites. But I think it is not correct because each site type has own WT mean level and WT seasonal dynamic. Anyway it must be mentioned in a paper text.

**Response:** Water table sensors were installed in a lawn about 1.5 m away from the EC raft. Unfortunately, it was impossible for us to measure the distance of WT to peat below as the bottom was formed by loose peat slurry. Therefore, we considered that the continuous WT logger was the most reliable source of data on the seasonal variation in WT.

**References**

Bao, Z., Okubo, T., Kubota, K., Kasahara, Y., Tsurumaru, H., Anda, M., Ikeda, S. & Minamisawa, K. (2014). MetaproteomiciIdentification of diazotrophic methanotrophs, and their tissue localization in field-grown rice roots. Applied and environmental microbiology, AEM00969.

Bosse, U. & Frenzel, P. (1997). Activity and Distribution of Methane-Oxidizing Bacteria in Flooded Rice Soil Microcosms and in Rice Plants (Oryza sativa), Appl. Environ. Microb., 63, 1199–1207.

Doronina, N. V., Ivanova, E. G., Suzina, N. E., & Trotsenko, Y. A. (2004). Methanotrophs and methylobacteria are found in woody plant tissues within the winter period. Microbiology, 73(6), 702-709.

Frenzel, P. & Rudolph, J. (1998). Methane emission from a wetland plant: the role of $CH_4$ oxidation in *Eriophorum,* Plant Soil, 202, 27–32.

Karofeld, E., Rivis, R., Tõnisson, H. & Vellak, K. (2015). Rapid changes in plant assemblages on mub-bottom hollows in raised bog: a sixteen-year study. Mires and Peat, 16. 11, 1-13.

King, G. M. (1994). Associations of methanotrophs with the roots and rhizomes of aquatic vegetation. Applied and Environmental Microbiology, 60(9), 3220-3227.

Larmola, T., Tuittla, E-S., Tiirola, M., Nykänen, H., Martikainen, P. J., Yrjälä, K., Tuomivirta, T. and Fritze, H. (2010). The role of *Sphanum* mosses in the methane cycling of a boreal mire, Ecology, 91, 2356–2365.

Peltola, O., Raivonen, M., Li, X., & Vesala, T. (2018). Technical note: Compariosn of methane ebullition modelling approaches used in terrestrial wetland models, Biogeosciences, 15, 937–951.

Wik, M., Crill, P. M., Varner, R. K., & Bastviken, D. (2013). Multiyear measurements of ebullitive methane flux from three subarctic lakes, J. Geophys. Res-Biogeo., 118, 1307–1321.

---

## Referee Comment (RC2) · Anonymous Referee #2 · 19 Apr 2019

The manuscript deals with an assessment of natural methane emission from a patterned boreal bog in southern Finland. The floating gas traps were used to estimate ebullition flux. The authors investigate observed methane fluxes with environmental parameters and ecosystem level methane flux from the chamber and EC methods. The paper contains some interesting material, very impressive introduction, is reasonably well written and is generally well referenced. In summary, the manuscript might be published after revision.

Specific comments

Line 22 – Median estimation of observed fluxes is more representative than a range of

variations. 58, 60, 69, 70, 73, 74, etc – please, check the absence of comma sign in references. i.e. (Strack et al. 2005) instead (Strack et al., 2005).

109 – Expand the climate description for the Siikaneva site, including, snow depth, freeze and ice depth, length of the frost-free period (growing degree days).

115 – Clarify the difference between the observation sites, depth of the studied pool in the central part and at the edge. See specific comment 2 form Reviewer 1. Your response should be added to the manuscript.

145 – Briefly describe how methane concentration from fresh ebullition was interpolated for weekly intervals.

153 - Is it possible to find ebullition flux from the moss cover surface using chamber observations? Potentially, gas bubbles can accumulate within porous peat and then goes up. It should be accounted for upscaling of methane ebullition flux.

176 – Were any significant differences between median and mean (average) ebullition fluxes? The distribution of observed fluxes is far from normal or Gaussian distribution fit (see A1), so median estimation is more representative than mean values.

335 – Methane ebullition flux is weakly related to peat temperature at the deep layer (see Fig. 3). The peat temperature at 50 cm has a seasonal maximum on 220-240 DOY, while CH4 flux has maximum earlier. What was the pool depth? Does temperature at pool bottom correlate with ebullition flux? What is the reason for the observed relation between methane flux and peat temperature at the depth of 5 cm? Is it the depth where the methane is generating? Discuss it.

590. Fig. 1-6 – I'd recommend to use traditional denote of the date (1-Aug) instead of a number of the day in the X-axis.

590. Fig 1-5 – Y-axis title is mg CH4 m-2 d-1, should be (CH4 flux, mg m-2 d-1)

---

## Author Comment (AC2) · 10 May 2019

Dear editors of the *Biogeosciences*

Revision for "Multiyear methane ebullition measurements from water and bare peat surfaces of a patterned boreal bog" (manuscript no. bg-2018-532).

We want to thank the two Referees for their constructive comments that certainly improved the report. Accordingly, we have revised our manuscript following the suggestions made by the Referee #1 and the Referee #2, or replied in detail why we have not followed their guidelines in some places. Here you will find the original statements of the referees and our responses to them.

**Referee #1:**

General comments

I found this paper well written and containing useful results. Design of the study is optimal for chosen object and helps to represent studied phenomena. Methods were described good enough. Especially I appreciate honest remarks on several methodological details. Literature was presented comprehensively, all important available studies were cited and used for comparison. It is very nice that photo of studied object was given in the paper (but in comments below I strongly recommend to give more photos to give a reader as much as possible information on your object). Nevertheless I recommend to improve this manuscript in several directions (they are described in specific comments section). After that I think this paper should be published.

Specific comments

**Comment:** Lines 54-55. It seems to me that in these lines it was pointed out that during methane transport in vascular plants methane is not consumed by methanotrophs. I think this is wrong because there are a lot of papers showing presence of methanotrophs in plant tissues (for example, Bao et al., 2014; Doronina et al., 2004; King, 1994). Am I right?

**Response:** Yes, this is true for several wetland species such as rice *Oryza sativa* (Bosse and Frenzel, 1997) and cattail *Typha latifolia* and *Calamagrostis canadensis* (King, 1994). However, in bogs so far significant methane oxidation has not been detected in vascular plants, such as *Eriophorum angustifolium* and *E. vaginatum* (Frenzel and Rudolph, 1998), but in *Sphagnum* mosses (Larmola et al. 2010). We will included this in the Introduction.

**Changes in manuscript:** Lines 55–58: We added a sentence "It is known that part of methane can be oxidized also in plants, such as rice (Bosse and Frenzel, 1997), but so far significant methane oxidation has not been detected in bog plants, such as *Eriophorum* spp. (Frenzel and Rudolph, 1998)."

**Comment:** Lines 110-115. I think your object should be described in paper text more comprehensively. I understand that there are a lot of papers about Siikaneva station peatland. But more information important for YOUR study should be given. In general you describe your research sites as peatland. But you also compare methane fluxes from your site with fluxes

from ponds and lakes. It seems to me that there is an ambiguity. Do you think that your objects are something between shallow lake and peatland?

It is very important for future reader to understand what EXATLY is your object. That is why more information on factors of methane emission should be given in the paper text. What was the water depth in all three site types (OW, EW and BP)? Why bare peat is bare? It is not typical for natural intact wetlands in Canada or Russia, where water table depth is about 0 cm or higher and moss cover is continuous. Why these parts of peatland are so wet (or so submerged)? Does peat on your sites removed by erosion (or by any secondary process, but not in inherent peatland development)? If so it decreases methane emission (because relatively young and rich in substrates peat layer was removed). Or your sites are in an inner water channel (stream) inside the peatland?

I strongly recommend to add photos of bare peat surface and water's edge (EW in your terms) sites and probably small map of your peatland to see where your sites are situated.

**Response:** Our study site is a patterned peatland with patches without moss cover. Those patches are either shallow with visible peat surface having their water level at the surface or deeper, covered by open water. In a way, our objects are something between very small and shallow lake and peatland.

Water depth in BP in 2014 was on average -1.8 cm. Water depth in the pools is not easy to measure as it is hard to determine what is the bottom of the pools: on average there is one meter of water over very loose peat slurry. The water depth does not generally differ between EW and OW as these peatland pools do not have shallower banks, but are deep right away at the edge of the surrounding moss cover. The shallow arctic lakes in Wik et al. (2013) have organic sediment layers and their water depth varies around 0-6 m making them not that different from our pools except being much larger in their area.

The bare peat surfaces or 'mud-bottom hollows' found in our site are commonly described in boreal peatlands in Estonia, Finland, Sweden, Russia, Western Europe and North America (Karofeld et al., 2015 and the references therein). Their formation processes are not well known but according to Karofeld et al. (2015) at least in this region they are not formed due to erosion but are inherent part of peatland development.

We have included a more detailed description of our study objectives and the aerial photo of the site (Figure below) to Methods.

[Figure]

Figure: Aerial photo of the study site in Siikaneva bog. Red lines with dots mark the floating gas traps in open water (OW) and water's edge (EW). Red circles mark the area within what the gas traps were placed on bare peat surfaces (BP) that are seen as brownish-grey in the photo. The eddy covariance (EC) raft is marked with the red x.

**Changes in manuscript:** Lines 126-130: We have added "BP are patches of visible peat that have WT at or near surface. For example in 2014, WT in BP was on average -1.8 cm. W is without a clear bottom but have on average one meter of water over very loose peat slurry and their water area starts directly from the edge of the surrounding moss cover. As it is difficult to determine what is the bottom of the pools, we did not measure the water depth or temperature in the bottom of the W."

**Comment:** Lines 127-128. How do you define where is open water (OW) and where is water's edge (EW)? Based on water depth? It is important because methane emission is known to be WT-dependent (see Wik's papers from your reference list for example).

**Response:** EW and OW were not defined based on the water depth but simply by the distance to the nearest vegetated surface. EW was right next to the moss and OW was the furthest away from it in the middle of the pool (at least 1m, on average about 3 m away from the edges). This separation was chosen to reflect the potential difference in availability of substrate for methanogenesis.

**Changes in manuscript:** Lines 137-140: We have modified the sentence "To study the potential difference in availability of substrate for methanogenesis some gas traps were anchored further away from the surrounding moss cover at the center of the pools (open water, OW), while the other gas traps were anchored at the water's edge (EW) right next to the moss (Fig. x).".

**Comment:** Lines 137-139. I have several questions to discuss on using triggered ebullition gas concentration for calculation of the total emission.

**a)** As I understand bubbles release to the surface from the peat when methane concentration in them reaches a certain threshold. If it is right, triggered ebullition gas concentration is lower than "real" ebullition gas concentration because gas concentration in triggered bubbles did not reach a certain threshold. Hence methane and CO2 concentrations and their fluxes are underestimated using your methodology (not by much I think). If I am right please mention it in the paper text.

**b)** Methane is poorly dissolved in water. And concentration of dissolved methane in peatland water is usually high and close to saturation level. That's why I think that methane concentration in funnels during a week (actually it is less than a week, because bubbles do not release right after funnel installation) can be more or less constant and not decreased by diffusion. Did your compare methane concentration in funnels after week of exposition and methane concentration after triggered ebullition in the end of this week? I have never read about methodology you used and think that it is novel. Any novel methodology should be assessed. For example, Martin Wik (see papers cited in your manuscript) use methane concentration in funnels after week (or couple of weeks) of exposition. It is always risky to use not in situ concentration in such heterogeneous environments as peatlands.

**Response:**

**a) )** Bubbles are not released from the peat after reaching certain methane concentration but depending on the bubble volume (Peltola et al., 2018). Volume in turn is controlled by pressure and temperature that affect methane solubility.

**b)** We have measured methane concentration of the gas caught in the funnels in 2014 and compared those with the fresh samples triggered from the peat. The concentrations in the funnels were clearly lower than in the fresh ebullition samples (see the table below). Therefore, after 2014 we did not measure concentrations from the funnels anymore but only took fresh bubble samples to get more accurate concentrations. We will include this in the methods; Table has been added as an appendix (A1).

| Date | **Mean CH4 concentration ml/l** | | | |
| --- | --- | --- | --- | --- |
| | **W funnel** | **BP funnel** | **W fresh** | **BP fresh** |
| 10-Jun | 17 | 45 | 293 | 268 |
| 16-Jul | 61 | *245* | 420 | 313 |
| 13-Aug | 86 | 152 | 399 | 172 |
| 05-Sep | 49 | *63* | 379 | 221 |

*Numbers in Italic indicate the concentration in a single measured funnel, not a mean of many funnels*

**Changes in manuscript:** Lines 147-150: We have added "Methane concentrations of the gas caught in the gas traps during the weekly sampling periods were measured in 2014 and compared with methane concentration of fresh ebullition samples. We found methane concentrations in the gas traps to be clearly lower than in the fresh ebullition samples (Table A1), and thus methane concentration of the gas caught in the traps was assumed to dilute during the weekly sampling periods due to diffusion. Therefore, methane…"

**Comment:** Line 158. Where exactly your water table sensor was placed, in what site? As I see on Figure 3 you use the same WT data for prediction of fluxes from all three types of sites. But I think it is not correct because each site type has own WT mean level and WT seasonal dynamic. Anyway it must be mentioned in a paper text.

**Response:** Water table sensors were installed in a lawn about 1.5 m away from the EC raft. Unfortunately, it was impossible for us to measure the distance of WT to peat below as the bottom was formed by loose peat slurry. Therefore, we considered that the continuous WT logger was the most reliable source of data on the seasonal variation in WT.

**Changes in manuscript:** Line 175: We added in the sentence "…data loggers installed in a lawn about 1.5 m away  from the EC raft".

**Referee #2:**

The manuscript deals with an assessment of natural methane emission from a patterned boreal bog in southern Finland. The floating gas traps were used to estimate ebullition flux. The authors investigate observed methane fluxes with environmental parameters and ecosystem level methane flux from the chamber and EC methods. The paper contains some interesting material, very impressive introduction, is reasonably well written and is generally well referenced. In summary, the manuscript might be published after revision.

Specific comments

**Comment:** Line 22 – Median estimation of observed fluxes is more representative than a range of variations.

**Response:** This is a good point. We have now added total flux medians for each year in the abstract.

**Changes in manuscript:** Lines 22-24: We have added "…with median 2 mg m$^{-2}$d$^{-1}$, … with median  3 mg m$^{-2}$d$^{-1}$, … with median 28 mg m$^{-2}$d$^{-1}$…" in the sentence.

**Comment:** 58, 60, 69, 70, 73, 74, etc – please, check the absence of comma sign in references. i.e. (Strack et al. 2005) instead (Strack et al., 2005).

**Response:** We have now carefully checked and corrected all the comma signs in references.

**Changes in manuscript:** Lines 62, 64, 73, 74, 77, 78, 84-88, 99, 121, 168, 269, 285 and 295 comma sign was added in references.

**Comment:** 109 – Expand the climate description for the Siikaneva site, including, snow depth, freeze and ice depth, length of the frost-free period (growing degree days).

**Response:** We have now added in the site description the 20-year average snow depth in March that is the month with the thickest snow cover. We have also added the 30-year average length of growing season. Unfortunately, we do not have data of freeze and ice depths in the area. However, Siikaneva peatland complex is situated within the southern boreal vegetation zone, where there is no permafrost. The soil has generally melted in the beginning and starts freezing after the end of growing season, which is when we have conducted all our measurements.

**Changes in manuscript:** Lines 110-111: We have added "…the snow depth in March (with the thickest snow cover) is 36 cm,". Line 114: We have added "…the average length of growing season is 168 days,".

**Comment:** 115 – Clarify the difference between the observation sites, depth of the studied pool in the central part and at the edge. See specific comment 2 form Reviewer 1. Your response should be added to the manuscript.

**Response:** We have now added more detailed descriptions of BP and W in the manuscript and addressed the pool depth question. Please, see our answer above to the comment 2 from Referee #1.

**Changes in manuscript:** Lines 126-130: We have added "BP are patches of visible peat that have WT at or near surface. For example in 2014, WT in BP was on average -1.8 cm. W is without a clear bottom but have on average one meter of water over very loose peat slurry and their water area starts directly from the edge of the surrounding moss cover. As it is difficult to determine what is the bottom of the pools, we did not measure the water depth or temperature in the bottom of the W." Lines 137-140: We have modified the sentence "To study the potential difference in availability of substrate for methanogenesis some gas traps were anchored furthest away from the surrounding moss cover at the center of the pools (open water, OW), while the other gas traps were anchored at the water's edge (EW) right next to the moss (Fig. 2)."

**Comment:** 145 – Briefly describe how methane concentration from fresh ebullition was interpolated for weekly intervals.

**Response:** We used linear interpolation to assess methane concentration of each measurement day based on the average methane concentrations of fresh ebullition samples from different W and BP. Methane concentrations were interpolated separately for W and BP with the formula

$$f = f_0 + [(t - t_0) / (t_1 - t_0)] (f_1 - f_0)$$

where methane concentration $f$ in the day $t$ is calculated based on the known methane concentration $f_0$ in the day $t_0$ and another known methane concentration $f_1$ in the day $t_1$.

**Changes in manuscript:** Line 159: We have added 'linearly' to the sentence "Average methane concentration was interpolated linearly from the fresh ebullition samples for each weekly measurement day."

**Comment:** 153 - Is it possible to find ebullition flux from the moss cover surface using chamber observations? Potentially, gas bubbles can accumulate within porous peat and then goes up. It should be accounted for upscaling of methane ebullition flux.

**Response:** Sudden increases of methane concentration during chamber measurements can be assumed ebullition. We excluded these changes from our chamber flux data from 2014. We now went through all the chamber measurements and estimated that ebullition occurred twice in 210 measurements on moss cover surfaces, i.e. in 0.8 %. Due to this there is a good reason to assume that ebullition from vegetated surfaces would not greatly contribute to the total flux in the bog site.

**Changes in manuscript:** Lines 333-335: We have added the sentence "We estimated ebullition to occur only twice in 210 measurements on moss cover surfaces, i.e. in 0.8 %, of our 2014 chamber data."

**Comment:** 176 – Were any significant differences between median and mean (average) ebullition fluxes? The distribution of observed fluxes is far from normal or Gaussian distribution fit (see A1), so median estimation is more representative than mean values.

**Response:** We fully agree that median values are more representative of ebullition fluxes than mean values. We have now changed the means to medians, as we should have realized to report them already in the first place. Median ebullition fluxes were slightly lower than mean fluxes as can be seen in the table below.

|      | Means   | Medians |
|------|---------|---------|
| 2014 | 1 - 90  | 0 - 57  |
| 2015 | 0 - 41  | 0 - 33  |
| 2016 | 14 - 71 | 10 - 67 |

**Changes in manuscript:** Line 192: We have added "…with medians 2, 3 and 28 mg m$^{-2}$d$^{-1}$…" Lines 193-194: We have modified the sentence to be "Weekly medians of individual gas traps were 0–57 mg m$^{-2}$d$^{-1}$ in 2014, 0–33 mg m$^{-2}$d$^{-1}$ in 2015 and 10–67 mg m$^{-2}$d$^{-1}$ in 2016." Lines 234-235: We have changed the sentence to report medians: "…weekly median of ebullition for different surfaces ranged from 0 (measured from BP in 2014) to 37 mg m$^{-2}$ d$^{-1}$ (measured from OW in 2016)".

**Comment:** 335 – Methane ebullition flux is weakly related to peat temperature at the deep layer (see Fig. 3). The peat temperature at 50 cm has a seasonal maximum on 220-240 DOY, while CH4 flux has maximum earlier. What was the pool depth? Does temperature at pool bottom correlate with ebullition flux? What is the reason for the observed relation between

methane flux and peat temperature at the depth of 5 cm? Is it the depth where the methane is generating? Discuss it.

**Response:** As we have explained above, it is hard to determine what the bottom of the pool is (please, see again our answer the comment 2 of Referee #1). Therefore, we have not measured pool depth or temperature at pool bottom. Methane flux has the same relation with peat temperature in each depth. Although we have all the reasons to assume that methane is evolving in the deeper layers, ebullition correlated strongest with temperature in 5 cm (in the model with 5 cm T AIC-value was the lowest). However, the difference to relation of peat T in other depths was not large and the temperatures within peat profile are strongly intercorrelated.

**Changes in manuscript:** No changes in the manuscript.

**Comment:** 590. Fig. 1-6 – I'd recommend to use traditional denote of the date (1-Aug) instead of a number of the day in the X-axis.

**Response:** We have now changed the X-axis of the figures to display dates instead of day of year.

**Changes in manuscript:** Lines 609-620: X-axis in the figures 3-6 has been changed to display the date as d/m/yy, for example "1/5/14".

**Comment:** 590. Fig 1-5 – Y-axis title is mg CH4 m-2 d-1, should be (CH4 flux, mg m-2 d-1).

**Response:** This is a good point. We have now corrected the Y-axis titles in the figures.

**Changes in manuscript:** Lines 609-620: Y-axis in the figures 3-6 has been changed to "$CH_4$ flux, mg m$^{-2}$ d$^{-1}$".

We hope that the editors and the Referees find the quality of our revised manuscript sufficient to warrant publication in the *Biogeosciences*. We thank the Associate Editor and the Referees for their valuable and constructive comments.

On behalf of co-authors,
Sincerely yours,

Elisa Männistö

School of Forest Sciences, University of Eastern Finland, P.O. Box 111, FI-80101 Joensuu, Finland
Tel.: 358-400 639608
E-mail: elisa.mannisto@uef.fi

**References**

Bao, Z., Okubo, T., Kubota, K., Kasahara, Y., Tsurumaru, H., Anda, M., Ikeda, S. & Minamisawa, K. (2014). MetaproteomiciIdentification of diazotrophic methanotrophs, and their tissue localization in field-grown rice roots. Applied and environmental microbiology, AEM00969.

Bosse, U. & Frenzel, P. (1997). Activity and Distribution of Methane-Oxidizing Bacteria in Flooded Rice Soil Microcosms and in Rice Plants (Oryza sativa), Appl. Environ. Microb., 63, 1199–1207.

Doronina, N. V., Ivanova, E. G., Suzina, N. E., & Trotsenko, Y. A. (2004). Methanotrophs and methylobacteria are found in woody plant tissues within the winter period. Microbiology, 73(6), 702-709.

Frenzel, P. & Rudolph, J. (1998). Methane emission from a wetland plant: the role of $CH_4$ oxidation in *Eriophorum,* Plant Soil, 202, 27–32.

Karofeld, E., Rivis, R., Tõnisson, H. & Vellak, K. (2015). Rapid changes in plant assemblages on mub-bottom hollows in raised bog: a sixteen-year study. Mires and Peat, 16. 11, 1-13.

King, G. M. (1994). Associations of methanotrophs with the roots and rhizomes of aquatic vegetation. Applied and Environmental Microbiology, 60(9), 3220-3227.

Larmola, T., Tuittla, E-S., Tiirola, M., Nykänen, H., Martikainen, P. J., Yrjälä, K., Tuomivirta, T. and Fritze, H. (2010). The role of *Sphanum* mosses in the methane cycling of a boreal mire, Ecology, 91, 2356–2365.

Peltola, O., Raivonen, M., Li, X., & Vesala, T. (2018). Technical note: Compariosn of methane ebullition modelling approaches used in terrestrial wetland models, Biogeosciences, 15, 937–951.

Wik, M., Crill, P. M., Varner, R. K., & Bastviken, D. (2013). Multiyear measurements of ebullitive methane flux from three subarctic lakes, J. Geophys. Res-Biogeo., 118, 1307–1321.